

# Assessment of shallow landslide susceptibility using an artificial neural network in Enshi region, China

Bin Zeng[1,*], Wei Xiang[2], Joachim Rohn[3], Dominik Ehret[4], Xiaoxi Chen[1]
1. School of Environmental Studies, China University of Geosciences, Wuhan 430074, Hubei, China
2. Faculty of Engineering, China University of Geosciences, Wuhan 430074, Hubei, China
3. GeoZentrum Nordbayern, Friedrich-Alexander-Universität Erlangen-Nürnberg, Erlangen 91054, Germany
4. Dept. 95 - State Engineering Geology, State Office for Geology, Resources, and Mining, 79104 Freiburg i. Br.,
Germany
*Corresponding author:* Bin Zeng, Ph.D.
Affiliation: School of Environmental Studies, China University of Geosciences.
Affiliation address: No. 388 Lumo Road, Wuhan, Hubei, 430074, P.R. China.
Email: zengbin_19@126.com. Tel: 86-27-67883473. Fax: 86-27-87436235.



**ABSTRACT**
Landslides are one of the most common and damaging natural hazards in mountainous areas.
However, due to the complex mechanisms that influence the activation of landslides, it can often be
very difficult to predict exactly when a landslide will occur. Therefore, research on landslide
prevention and mitigation mainly focuses on the distribution forecasting of unstable slopes that are
prone to landslides in specific regions and under multiple external forces. The prediction of the spatial
distribution of these unstable slopes, termed Landslide Susceptibility Zonation, is important in helping
with government land-use planning and in reducing unnecessary loss of life and property. Researching
unstable slopes in the Silurian stratum in Enshi region, China, this investigation established a GIS and
artificial neural network (ANN)-based method to predict the distribution of potential landslides in this
area. Based on the failure mechanism analysis of typical landslides in Silurian stratum, development
of evaluation index system which represents the most relevant factors that influence the slope stability,
and establishment of intelligent slope stability susceptibility prediction model by artificial neural
network, the spatial distribution of unstable slope zones that are prone to landslides were predicted in
the study area. The results were further well supported from remote sensing data and field
investigations. This research proves that the spatial unstable slope prediction method based on
intelligence theory and GIS technology is accurate and reliable.



## 1 Introduction

Slope failure is a complex natural hazard problem which causes damage to property and loss of life
in almost every country. To remedy this, it is necessary to scientifically assess areas that are susceptible
to landslide, and so substantially to decrease the damage caused by landslides (Lee et al., 2004). Due to
the variety of influencing factors and nonlinear physical processes involved in landslides, it is difficult
to predict the potential instability of slope regions where a landslide has not yet occurred.
In order to assess the potential for landslide, it is a requirement to identify and analyze the
influencing factors first (Lee and Evangelista, 2006). Depending on different landslide types and failure
mechanisms, the number of influencing factors was normally decided in the range of 4 to 15 (Shan et al.,
2002; Sezer et al., 2011; Pourghasemi et al., 2012; Akgun et al., 2012; Kayasha et al., 2013). Moreover,
some researchers analyzed the necessity and weight of different influencing factors (Pradhan and Lee,
2010; Ozdermir and Altural, 2013; Mahalinggam et al., 2016), and some researchers analyzed the best
forecasting accuracy due to combination of different number of influencing factors (Pradhan and Lee,

2010).

Afterwards an appropriate and targeted prediction model is the key to the success of landslide
susceptibility assessment. To date, a number of different methods combined with GIS have been





developed to assess slope stability susceptibility: frequency ratio model and weighting factor method
(Yalcin et al, 2011), analytical hierarchy process method (Wang and Yi, 2009), statistical analysis
method (Lee and Min, 2001), logistics regression method (Dai and Lee, 2002; Devkota et al., 2012),
multi-criteria decision method and likelihood ratio method (Akgun, 2012), evidential belief function
model (Althuwaynee et al., 2012), spatial multi-criteria evaluation model (Pourfhasemi et al., 2014),
fuzzy overlay method (Kirschbaum et al., 2016), heuristic approach (Ruff and Czurda, 2008), grey
clustering method (Zhang et al., 2009), neuro-fuzzy model (Sezer et al., 2011; Bui et al., 2012), and
artificial neural network method (Wang et al., 2005; Ercanoglu, 2005; Ermini et al., 2005; Gómez and
Kavzoglu, 2005; Lee et al., 2006; Pavel et al., 2007; Wang et al., 2016). Some researchers also
compared the predictive capability of different models in the same area and provided recommendations
of a more appropriate model (Kanungo et al., 2006; Yalcin et al, 2011; Akgun, 2012; Ozdermir and
Altural, 2013; Althuwaynee et al., 2014; pourfhasemi et al., 2014; Goetz et al., 2015; Wang et al., 2016).
However, no general agreement has yet been reached about the best method for landslide susceptibility
assessment, all known methods have their advantages and disadvantages (Ercanoglu, 2005); but
utilization of intelligent method has become more commonly used in recent years.
The key problem with the spatial predictions of landslide risk is the establishment of a prediction
model that is consistent with the failure mechanism of landslides (Yin, 1992), while the forecast



method and criteria are at the core of a forecast model. A large amount of research has been conducted
in regards to the spatial prediction of landslides, but there are still some points that can be improved.
First, some of the results from the landslide prediction model include the distributions of landslides
that have already occurred. However, in these areas the stress accumulated in the slope may have been
released through the sliding deformation of the slope, reducing the entire potential energy of the slope
to such an extent that these areas are temporarily stable. The real danger lies in slopes that have not
yet slid because these could develop into potentially unstable slopes, or even landslides, triggered by
various external forces. Therefore, the focus of research to predict the spatial distribution of landslides
lies on studying these types of potentially unstable slopes. Secondly, the slope itself is a complex
nonlinear, anisotropic body and the external forces are in a non-steady state that is constantly changing
in space. Thus, because of the different failure mechanisms of landslides, a targeted mathematical
model with the reasonable combination of valuation index system is required for more accurate
predictions. Lastly, the prediction of a landslide should include qualitative, quantitative, and
experiential factors. Expert knowledge and experience plays a very important role in these
predictions, so the combination of expert knowledge and technology will be central to future research
on the spatial predictions of landslides.

This paper focuses on the distribution of unstable slope zones in the Silurian stratum in Enshi



region, China. Based on the failure mechanism analysis of landslides in this specific area, a targeted
evaluation factor index system was developed, and a prediction model combined GIS technology with
artificial neural network was established for predicting distribution of unstable slope zones that are
prone to landslides in the study area. Compare with existing studies, some different attempts have been
carried out: (1) the research focused on the distribution of unstable slope zones rather than the existing
landslides, since the unstable slopes are more dangerous; (2) this research predicted the unstable slope
distribution only in Silurian stratum so as to avoid the interference due to differences in slope failure
mechanisms; (3) a "slope structure thematic map" was taken into account to better represent the especial
slope failure mechanism in Silurian stratum; (4) replaced the temporal variable of rainfall into a static,
spatial variable termed "catchment area" to better act as an influencing factor during the landslide
susceptibility. The research results can provide useful guidance for both landslide susceptibility
assessment and land planning processes.
**2 Background**
*2.1 Geological setting*

Enshi is located in the mountainous, southwestern area of the Hubei Province in China, and the

study area is located west-north of Enshi region, as shown in Fig.1.

For this study, only Silurian stratum (as shown in Fig.2) is chosen as the object stratum in the




research area. The benefits of this are clear: in the same stratum, the natural environment, geology and
hydrogeology condition, and particularly the failure mechanism, are all convergent, so that the
interference due to differences in slope failure mechanisms can be effectively avoided.
The study area is located in a mountainous area that ranged in elevation from 600 to 1200 m. The
surface water system is well developed; the rivers are generally between 20 and 40 m wide. There are
slopes on both sides of the valley and gully, generally with slope angles of 20º-40º. The valley is
V-shaped with a relative elevation from the bottom to the ridge between 200 and 600 m. Large ancient
debris deposits are distributed along the river and its tributaries.
The components of the Silurian stratum are mainly shale, mudstone, siltstone, silty mudstone,
silty clay, pelitic siltstone and other clasolites. The lithological strength is weak. Various external
geological effects, such as unloading, rainfall infiltration and erosion, and wind weathering, could
easily destroy the rock mass structure and provide a wealth of material for landslides. The main
components of loose soil are clay and silt clay with fragments. The main causes of formation are
eluvia, alluvia and colluvia. Generally, the thickness of loose soil is 0.5-10 m, with a maximum
thickness of 15 m. Due to the loose structure, and low physical and mechanical strength, the surface
loose soil can be easily turned into shallow landslide.
***2.2 Mechanism analysis of shallow landslides in the Silurian stratum***



---

2.2.1 Characteristics of landslides in the Silurian stratum
According to the filed investigation and statistic results, most of the landslides in Silurian stratum
in the study area are soil landslides (81.3%), with a volume of less than $10^6\,\mathrm{m}^3$ (81%), a slope angle
between 15° and 35° (82%), a thickness of less than 10 m (75%) or between 10 and 25 m (16%), and
they occurred during the rainy season from June to August (87.5%).
2.2.2 Geological and environmental conditions for the shallow landsides
The conditions causing shallow soil landslides (some examples in Fig.3) in the Silurian stratum are
related to the following factors. First, most of the slopes were between 10 and 200 m high and had free
surface in front. Second, most of the landslide bodies were weathered and loosely accumulated, with a
thickness of 2 to 10 m. Third, dip slope or skew slope (slope with a small angle between    slope surface
dip direction and stratum dip direction, as defined in Table 1) structure provided conditions of slide
surface. Fourth, the upper loose soil layer had weak permeability and strong water-holding capacity,
while the underlying bedrock was a relative aquifuge. Thus, the upper soil layer was easily soaked and
softened to form a slide surface near the interface of the soil layer and bedrock. Fifth, because the
mechanical strength of the soil was low, when the water content was increased, the shear strength ($\tau$)
decreased rapidly. Finally, long-term, heavy rainfall, and slope cutting were the main external factors
that triggered shallow soil landslides in the study area.



### 2.2.3 Evolution of shallow landslides in the Silurian stratum

Based on the analysis of the factors influencing the slope stability, a typical failure mechanism of

shallow landslides in Silurian stratum is described here. This process occurs after the accumulation

body has been formed on the bedrock. Often during rain events, a tension crack in the trailing edge of

the accumulation body appears at the top, extending to both sides. The tension fracture becomes a

major channel for surface water infiltration. In continuous rainfall conditions, there is also continuous

runoff of surface water and infiltration along the fracture channel. Water erosion washes out and

further loosens the soil along the fracture, which widens and deepens the cracks, extending the trailing

edge of the tension crack and gradually forming the sliding surface. However, due to the clay gravel

composition of the accumulation body, the infiltration of groundwater discharge is blocked. This

increase pore water pressure and decreases the effective stress of the accumulation body, especially

near the trailing edge of the cracks, which reduces the friction force between the particles. At the same

time, because the weight of the slope body has increased, the shear stress in the front edge of the

accumulation body gradually increases to a maximum intensity. The shear strength is gradually

reduced and the middle part of the slope body (locking section) reaches its peak strength. With further

deformation of the accumulation body, the strength of the soil mass is reduced, the sliding surface

joins, and the total anti-sliding force (shear strength) of the sliding surface nearly equals the total





sliding force until the landslide occurs.
**3 Methods**
***3.1 Establishment of an influencing factor system of the shallow landslide***
3.1.1 Screening of the index factors

Although there are several geological, topographical, and/or environmental parameters that can be

used to produce landslide susceptibility thematic maps, selection of these parameters depends on several
factors such as data availability, data quality, size of the study area, scale of the work, user experience
etc (Ercanoglu, 2005). For this study, according to the failure mechanism analysis of the shallow
landslides in Silurian stratum, except lithology factor, four most relevant landslide influence factors
(slope angle, slope structure, road influence and rainfall) were identified:

(1) Slope angle

The slope angle determines the distribution of the landslide based on the geometric features, and

also directly determines the stress distribution in the slope. Different slope angles not only affects the
magnitude of the residual stress along the existing or potential sliding surfaces but also determines the
form and mechanism of slope deformation failure. Controlled by topography, landslides in Silurian
stratum generally occurred in slopes with angles of 15°-35°, which accounted for approximately 82%
of all landslides in the same stratum.





(2) Slope structure
According to the field survey, most of the shallow landslides in Silurian stratum have occurred
either in dip slopes or in skew slopes (slope with a small angle between the slope surface dip direction
and stratum dip direction, as defined in Table 1). Because the sliding surface normally locates near the
separation plane of the loose soil layer and bedrock, when the slope aspect and dip direction of the
stratum are the same or intersect in a small angle, the sliding surface effect reflects more clearly.
(3) Road influence
The landslides in Enshi region are clearly influenced by human activities. The total number of
landslides and unstable slopes caused by human activities was 83% and mainly included slope cutting
during the construction of roads.
(4) Rainfall
Rainfall was the most active element that caused landslides in Silurian stratum. First, rainfall was
converted partly into surface runoff. Then, after the runoff continued for a long time, it formed a gully
on the surface that changed the surface morphology of the slope. After the gully was eroded deep
enough, it would provide the spatial conditions necessary for the deformation and failure of the slope.
In Enshi region according to field investigations, most of the landslide toes were distributed in stream
systems or gullies and the slope toe was constantly eroded which provided a free surface for the final





destruction of the slope.

Rainfall is a temporal variable, but for a spatial distribution prediction of landslides it should have

the characteristics of a spatial variable. Due to the importance of rainfall as an influencing factor, it is
necessary to change this factor from a temporal variable to a static evaluation index with spatial
distribution characteristic. Therefore, the concept of a "catchment area" is established. This represents
the capacity of slopes to collect surface water that has been transformed from rain. As shown in Fig.4,
using the catchment area concept, the temporal variable of rainfall is converted into a static spatial
variable and related to the slope stability.
3.1.2 Quantification of indices

The evaluation index is divided into qualitative and quantitative indices, which must be given a

quantitative value. In addition, the variables must all undergo compression processing or
non-dimensional data processing before they are used (Wang, 2000). Because the measurement units
of each variable are inconsistent, different variables have different levels of influence and some are
exaggerated, so it is necessary to eliminate the dimensional effect of the variables. In this study,
continuous variable indices, linear factor indices, and discrete variable indices are distinguished.
(1) Continuous variable index: slope angle

According to the difference principle, the relationship between landslide distribution density and





the slope angle in the study area are combined, and the slope angle is divided into four ranges, 0°-15°,
15°-35°, 35°-50°,50°-90°, so as to make the division of continuous variable states more reasonable,
and the prediction model more optimized.
(2) Linear factor indices: road influence buffer zone, stream system and gully influence buffer
zone
To deal with linear factors, it is necessary to determine the distance between landslides and these
types of factors, using the buffer analysis principle and statistical analysis to determine the radius of
influence. In addition, the minimum grid size for the spatial analysis, considered to be 50 m for
statistical analyses, is used. Considering the frequency of landslide distributions at different buffer
distances, the SC (Susceptibility Coefficient) method shown in Eq. (1) is adopted to obtain a
reasonable buffer distance.
$$SC_i = \ln\left(\frac{N_i}{A_i} \middle/ \frac{N}{A}\right) \qquad (1)$$

$SC_i$ – the sensitivity coefficient of a certain type in factor i (greater value indicates that a landslide
will occur easier in this section); $N_i$ – the number of landslides of a certain factor, i (type); $A_i$ – the
area (km$^2$) of a certain factor i; N – the total number of landslides in the study area; A – the total study
area (km$^2$).
According to this, the SC values representing the influence of the distance from stream systems





and roads can be calculated. The influence radius of the stream systems and gullies is 50 m and the
influence radius of the road is 100 m.

(3) Discrete variable index: slope structure

Based on the angle between the slope surface dip direction and stratum dip direction, the slope

structure can be divided into three main types: dip slope, reverse slope, and skew slope (Hoek & Bray,
1981).  Based on the filed investigation experience about slope stability, the slope stability from best
to worst is: reverse slope > skew slope > dip slope.

In the process of extraction, the stratum dip directions are divided into 8 ranges: 0°-45°, 45°-90°,

90°-135°, 135°-180°, 180°-225°, 225°-270°, 270°-315°, 315°-360°. Based on the stratum dip direction
and slope surface dip direction distribution maps calculated by ArcGIS, the angle between the slope
surface dip direction and the stratum dip direction is obtained by subtracting the superposition
calculation from the above two layers. The slope structure is divided into regions in the study area
according to Table 1.
3.1.3 Weight of index

The intelligent prediction system used in this paper is based on the neural network learning

memory sample rule that simulates the thinking in a human brain and automatically assigns weight
coefficients during the process of forecast calculations. The advantage of this computing process is



that it can effectively avoid the interference of human factors. However, the prediction system is based
on the thinking method of the human brain; therefore, the neural network method requires high
quantities of typical samples for correct learning and forecasting.
***3.2 Establishment of an artificial neural network prediction model***
3.2.1 Selection and prediction process of a neural network model

A BP (Back Propagation) neural network is a type of neural network that has a one-way

transmission of multilayer, feed-forward neural networks. A neuron is the basic unit of the neural
network. In the network, the neurons are arranged in layers that are composed of the input layer, the
hidden layer (of which there can be several) and the output layer. Because the prediction of unstable
slopes in theory is a process of functional approximations, then based on the influence factor
parameters in the input layer, the nonlinear mapping relationship with the corresponding output
parameters can be obtained (Zhang et al., 2005). Therefore, a BP neural network is very suitable for
addressing the prediction of unstable slopes.

The steps (flow-chart as shown in Fig.5) involved in using a BP neural network to predict the

distribution of unstable slope zones is as follows: First, typical and investigated unstable slopes are
used as the research objects. Various factors that may affect the stability of the slope are quantified to
be the input values of the input layer nodes. The stability state of these slopes is divided, quantified,




and regarded as the desired output of the output layer nodes. Then, the neural network is repeatedly
trained using these known samples until the total error of the network meets the precision requirement.
In this way, the network masters the relationship between the input factors and the expected output.
Finally, the input includes the influence factors of slope stability for unknown regions, and based on
the previously established and tested neural network prediction model, the spatial unstable slopes
distribution result for unknown region can directly obtained.
3.2.2 Construction of a BP neural network model

Determine the number of neurons for the input and output layers. Based on the previous

analysis of slope stability influence factors, the number of neurons in the input layer is 4 and the
number of neurons in the output layer is 1, which is the stability state of the slope.

Determine the number of hidden layers in the neural network. Any continuous function on a

closed interval can be approached by single hidden layer BP network. Thus, a three BP network can
complete any of n-dimensional to m-dimensional mappings (Feng et al., 2009). Therefore, the hidden
layer of this BP network is set to 1.

Determine the number of neurons in the hidden layer. According to the Kolmogrov theorem (Wen,

2004), given any continuous function $f: [0,1]n \rightarrow Rm$, f can be achieved through a three layer
forward neural network. The input layer has n neurons, the middle layer has $2n + 1$ neurons, and the



output layer has m neurons. Therefore, the neuron number of the hidden layer is: $2 \times 4 + 1 = 9$.
Determine the network training function. The "traingdx" function (momentum and adaptive
algorithm) can not only effectively avoids the problem of local minima but also adjusts the learning
rate. Therefore, it has high training efficiency and a stable training process (Wen, 2004), and is chosen
as the training function for the neural network.
Determine the initial weights and threshold values for the neural network. Based on a comparison
of different initial ranges, it is found that an initial value which is not too large has little impact on the
overall performance of the network, while a smaller initial range is more conducive to uniformly
random initial weights (Freeman, 1993). In the BP neural network, the initial network weights and
thresholds are given random values in the acceptable range. Finally, the structure of the designed BP
network is shown in Fig.6:
3.2.3 Pretreatment of sample data
The prediction principle of neural networks is that the neural network can effectively approximate
the inherent laws of the sample by studying and remembering the known samples, then carry out an
associated forecast according to the memory. Thus, the unknown samples to be predicted must be
similar to the known samples. At the same time, the known samples should also cover as much as
possible the different combinations of various factors to improve the forecasting ability of the network



(Zhang, 2006).
Sample data: 35 stable and unstable slopes in Silurian stratum in Enshi region are chosen as the
sample data, as shown in Fig.7. The recognition and mapping have been carried out by
geomorphological field survey. The defined unstable slopes are slopes that have deformation evidence,
and may prone to typical shallow soil landslides in Silurian stratum under multiple external forces. As
shown in Table 2, samples 1-25 are used for the network training and samples 26-35 are used to test
the performance of the network prediction.
Reprocessing of sample data: The response of neurons is between 0 (inhibition) and 1 (activation).
To ensure that the BP neural network learns as best as possible and to prevent small numerical
variables in the input from being overwhelmed by large numerical data, the sample data are
normalized before processing. Input values are converted to values between 0-1 with an appropriate
transformation. For the qualitative index, the ones and zeros of the binary logic calculations can be
used to express yes or no for the categories; for other indexes, the values were corresponding to
between 0.1 and 0.9 according to their contribution. As shown in Table 2.
3.2.4 Training and testing the forecasting ability of the established neural network model
(1) Training procedure
Based on MATLAB 7.1, the established neural network model was trained using samples 1-25:



The training code in MATLAB is as follow:
netgdx=newff(minmax(p1),[9,1],{'tansig', 'logsig'},'traingdx','learngdm','mse');
netgdx.trainParam.epochs=10000;
netgdx.trainParam.goal=0.00001;
netgdx =train(netgdx,p1,t1);
The training result is shown in Fig.8:
(2) Testing procedure
After the neural network had been trained and achieved the training goal, samples 26-35 were
used to test the predictive ability of the network. As shown in Table 3, the prediction accuracy for
samples 26-35 reached 80%. Therefore, the network was considered to have reached a stable state
with a good forecasting ability.
**4 Results and Discussion**
***4.1 Implementation of the spatial prediction for unstable slopes in the study area***
According to the analysis of main influencing factors based on slope failure mechanism in the
research area, four evaluation indices were taken into account during the prediction model, including:
slope angle, slope structure, influence buffer of streams and gullies, and influence buffer of roads.
Classification of the forecast unit in the study area: The spatial database of the thematic layer is

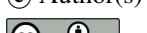



often vector data. When predicting the landslide susceptibility, it is necessary to convert the vector
data into raster data, which is the grid processing by ArcGIS. Based on topographical and geological
maps, the four thematic layers were gridded using 25 m x 25 m by ArcGIS. After gridding, the total
raster number in the study area was 81382, and in the actual calculations in Silurian stratum the total
area was approximately 103 km$^2$.

Creation of the impact factor thematic layer: the four parameters, which were used in the landslide

susceptibility in this study, were derived from remote sensing data, DEM and ancillary data from
fieldwork. The distribution of slope angle thematic map was calculated by the digital terrain analysis
function in ArcGIS; the distribution of road influence buffer and distribution of stream and gully
influence buffer thematic maps were calculated by the buffer analysis function in ArcGIS; for the
distribution of slope structure thematic map, firstly the distribution of slope aspect thematic map was
calculated by ArcGIS based on DEM data, then the distribution of bedrock dip direction thematic map
was gained according to geologic map and field investigation, finally the distribution of slope
structure thematic map can be obtained by overlying the above two thematic maps in ArcGIS. The
four thematic layers were obtained as shown in Fig.9.
*4.2 Distribution of unstable slope zones in the Silurian Stratum*

Based on a GIS superposition calculation of the four thematic layers, the predicted result of the



spatial distribution of unstable slopes in the study area was obtained, as shown in Fig.10.
As shown in Fig.10, the red areas are natural slopes which have not completely slid, but under the
continuing influence of external forces (i.e., rainfall, earthquakes, large-scale human engineering
activities, etc.) these areas will likely form shallow soil landslides in the Silurian stratum. In addition,
the potential instability region (red) in Fig.10 does not fully represent the exact boundaries of a single
slope that may slide in the future. If the slopes become unstable in the future, these regions will be the
origins of slope failure. Therefore, understanding the distribution of these regions is of great
importance for further determinations of the approximate ranges of unstable slopes.
The results also showed that the Neural Network Intelligent Forecasting System based on
repetitive learning and memory of a representative sample, under the premise of mastering the special
regularity of the Silurian stratum slope failure, was sufficient to dynamically assess weights for the
various factors. Although the weight of each impact factor is different, a landslide cannot be triggered
only by a single factor. Multiple factors in different combinations were given different weights by the
intelligence system, which not only improved the accuracy of predictions of regional slope stability
under complex topographic and geologic conditions but also avoided the errors associated with
subjective decisions about the weight of each factor.
***4.3 Accuracy verification of the spatial prediction model***




To further verify the accuracy of the intelligent prediction model, the predicted potentially
unstable region was verified by both remote sensing data (SPOT-5 satellite with resolution of 2.5 m)
and field investigations, as shown in Table 4.
The unstable slope zones determined by the result of the intelligent prediction are closely
reflected in the data determined from remote sensing and confirmations made through field
investigation. This proves that the intelligent prediction of unstable slopes based on a neural network
results in an accurate forecast, especially for shallow soil slope failure zones in Silurian stratum.
The good judgment exhibited by the intelligent forecasting system in regards to the influence of
slope gullies greatly improves the accuracy of the prediction. It is found from the predicted results that
most of the unstable slope zones, either at the toe or on sides of the border, are distributing along
continuous gullies that provide a free surface for slope deformation and failure. This is consistent with
the failure mechanism of landslides in Silurian stratum. It also highlights the benefit of stream and
gully distribution maps based on the catchment area concept. Compared with the traditional single
stream distribution map, the model is able to judge the distribution of the gullies in non-perennial
water areas, and these gullies have played a very important role in the occurrence of Silurian stratum
landslides.





As for results of the intelligent spatial prediction of unstable slopes, it cannot completely
represent the precise scope of each potentially unstable slope body and what is more significant is the
information about the initial damage site of unstable slope body. Therefore, in actual use, the predicted
potential instability region should also be combined with field surveys to determine the accurate range
of unstable slopes which would be destroyed by external forces in the future.
**5 Conclusions**
This paper established a relatively complete method for the spatial prediction of unstable slope
zones in the Silurian stratum in Enshi region that used slope failure mechanism analysis, GIS-based
data collection, evaluation index system development, ANN-based intelligent unstable slope
prediction model design. According to the results of remote sensing data and field investigation, the
prediction model is accurate and reliable. This method will be useful for the prediction of similar slope
disasters in mountainous area.
The study also made the following different attempts compare with other researches: (1) selected
a single Silurian stratum as study object to effectively avoid errors in forecast accuracy due to
different slope failure mechanisms in different strata; (2) focused on the prediction of  spatial
distribution of unstable slopes rather than existing landslides, since unstable slopes are much more
dangerous; (3) established the concept of "catchment area", so that rainfall can be indirectly



considered as a static, spatial evaluation index associated with slope stability, the use of "catchment
area" is also able to accurately describe the gully distribution which plays a crucial role in slope
stability; (4) established a slope structure thematic map which well reflects the specific failure
mechanism of shallow soil landslides in Silurian stratum.
**Author Contributions**
Bin Zeng contributed to data analysis and manuscript writing; Wei Xiang, Joachim Rohn and
Dominik Ehret proposed the main structure and key idea of this study; Xiaoxi Chen performed the
ANN calculation. All authors read and approved the final manuscript.
**Acknowledgments**
This work was supported by the open fund from Three Gorges Research Center for geo-hazards,
Ministry of Education, China [grant numbers: TGRC201007]; the national study abroad fund for
construction of high level university, China [grant number: 20073020].
**Conflicts of Interest**
The authors declare that they have no conflict of interest.

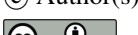



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



**Figure captions**

**Fig. 1.** The location of the study area.

**Fig. 2.** Generalized geological map. Only Silurian stratum is chosen as the study area.

**Fig. 3** Typical shallow soil landslides in the Silurian stratum in Enshi region.

**Fig. 4** Conversion of the temporal variable of rainfall into a static, spatial variable termed "catchment area" to better act as an influencing factor during the slope stability susceptibility

**Fig. 5** Flow-chart methodology for the prediction of unstable slope distribution in the Silurian stratum in Enshi region based on BP neural network.

**Fig. 6** The structure of the BP neural network model for the prediction of unstable slope zones in Silurian stratum

**Fig. 7** Stable and unstable slopes inventory map of the study area. The recognition and mapping have been carried out by geomorphological field survey.

**Fig. 8** Training procedure and error curve of the neural network model based on "traingdx" training function. During the training process, the error curve had good convergence and met the training goal within setting epochs, which means the "traingdx" training function can make the model has a stable training process and expected training result.

**Fig. 9** (a) Distribution of slope angle; (b) Distribution of road influence buffer; (c) Distribution of





stream and gully influence buffer; (d) Distribution of slope structure.

**Fig. 10** Results of the slope stability susceptibility assessment performed by the BP neural

network in the Silurian stratum in Enshi region.




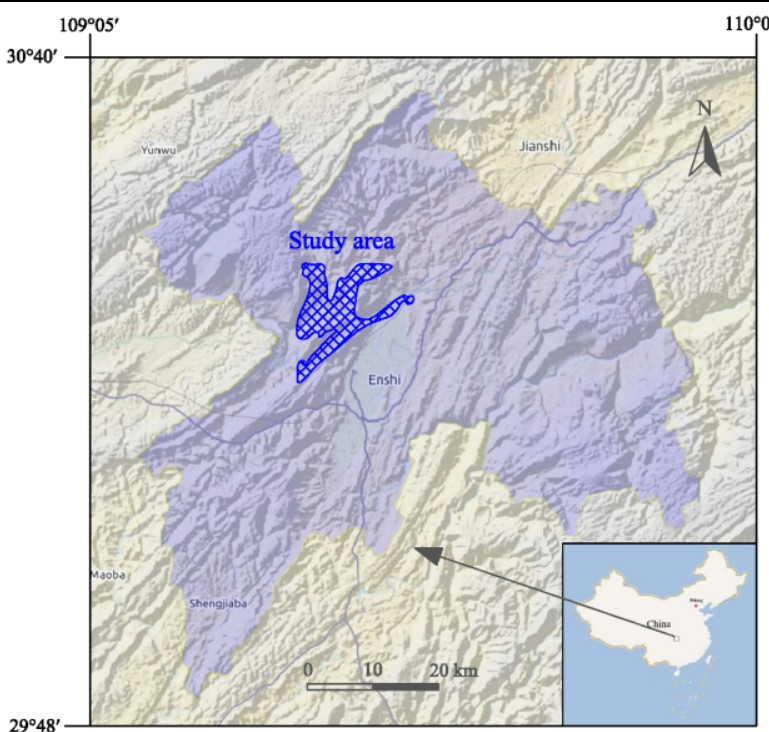

**Fig.1**



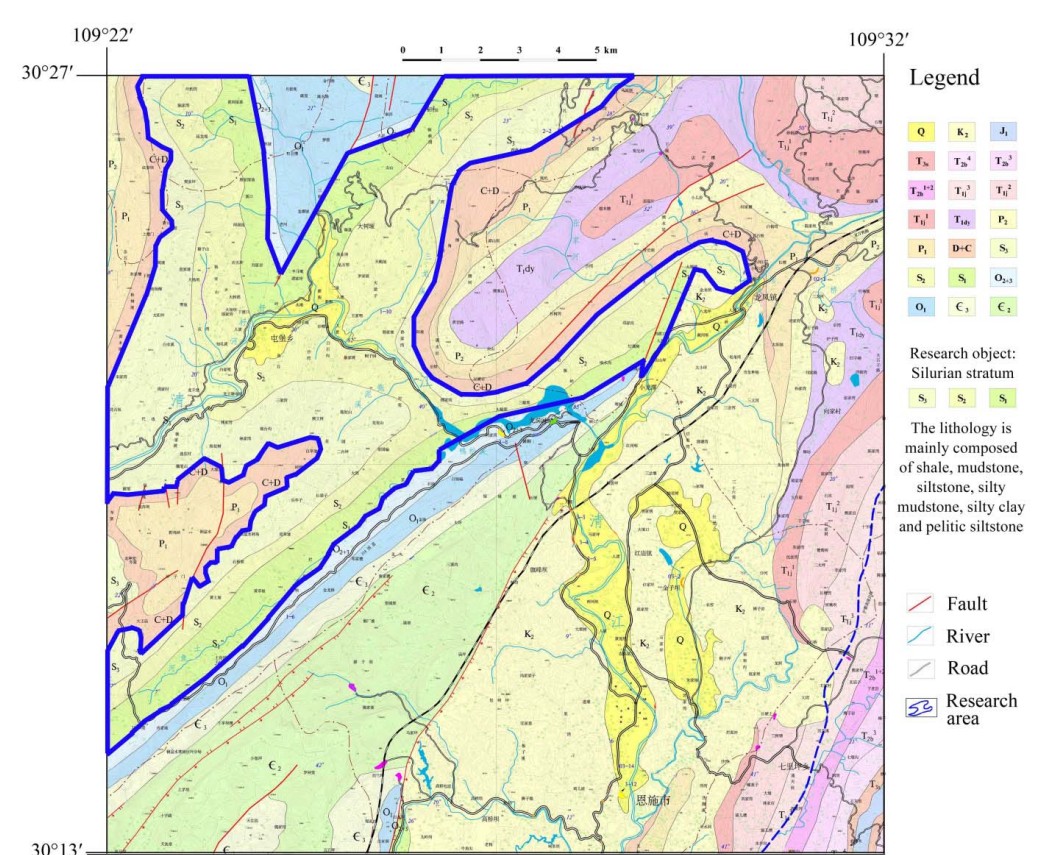

**Fig.2**



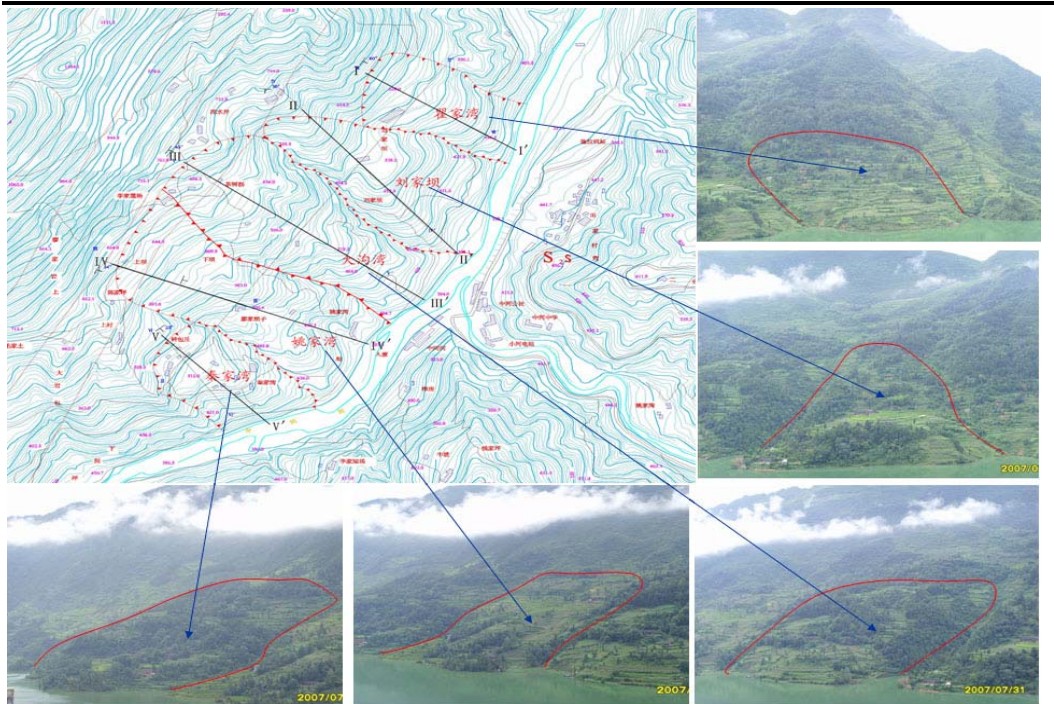

**Fig.3**





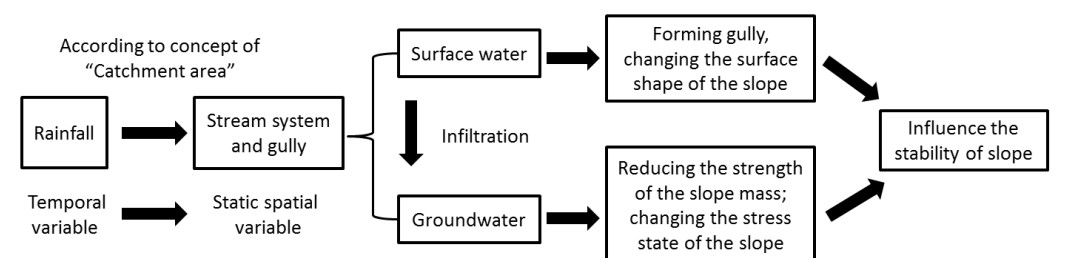

**Fig.4**

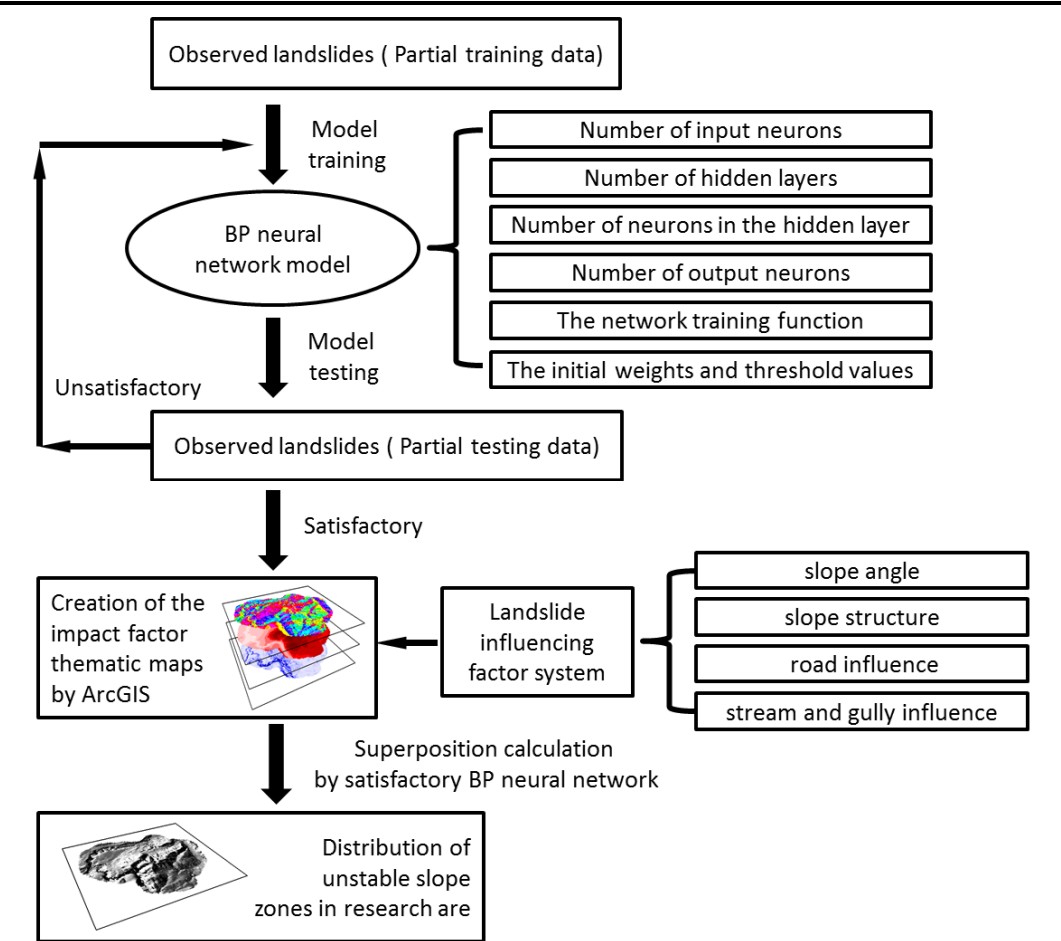

**Fig.5**



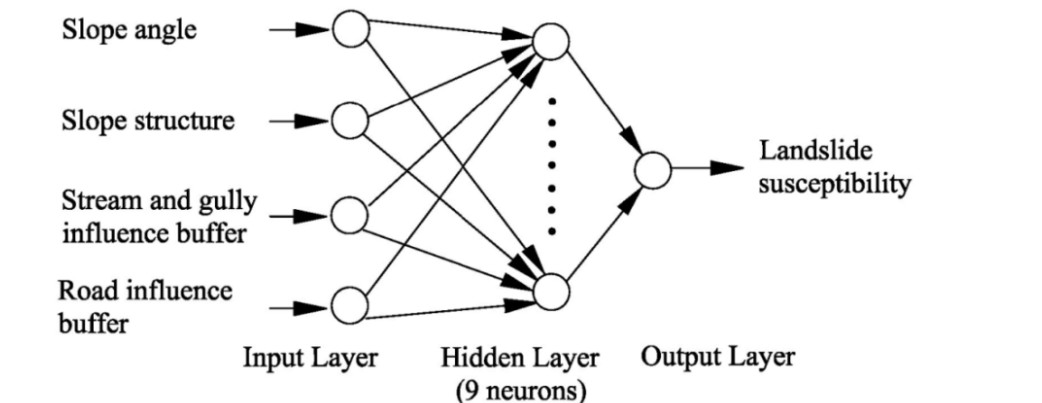

**Fig.6**



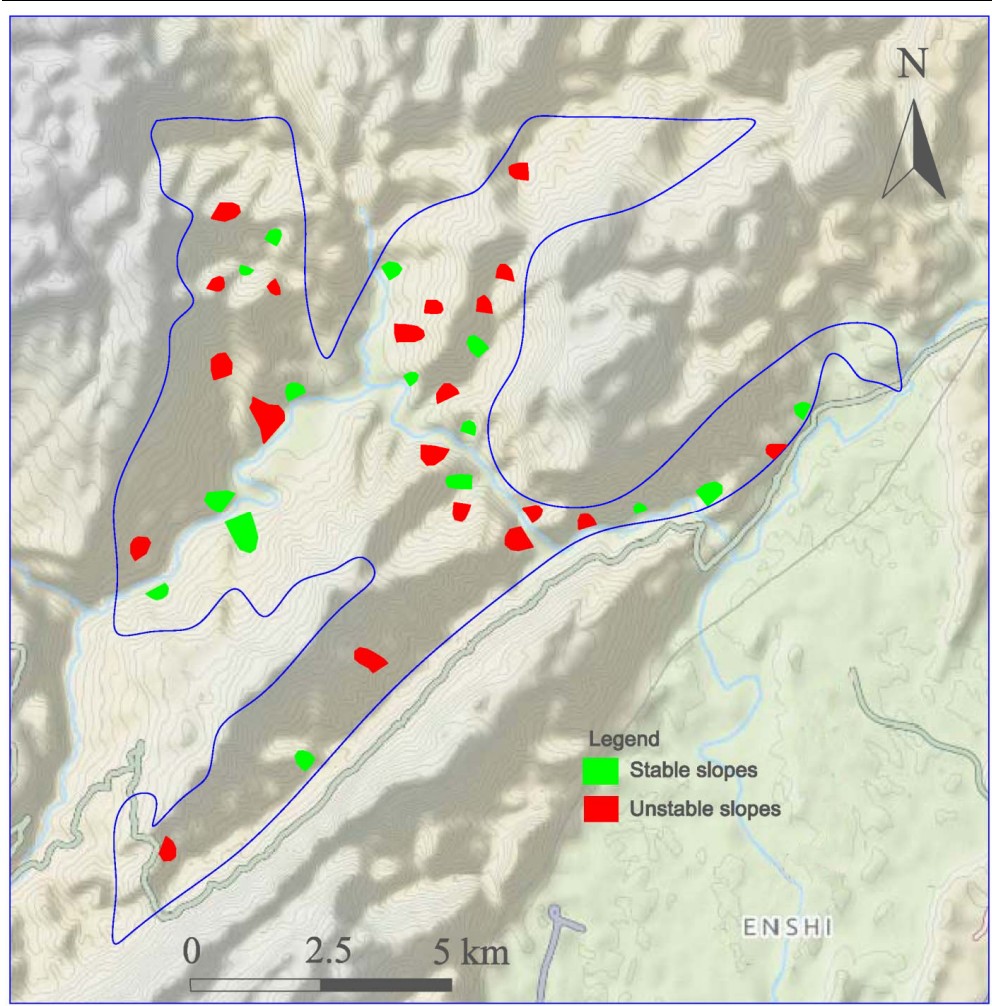

**Fig.7**




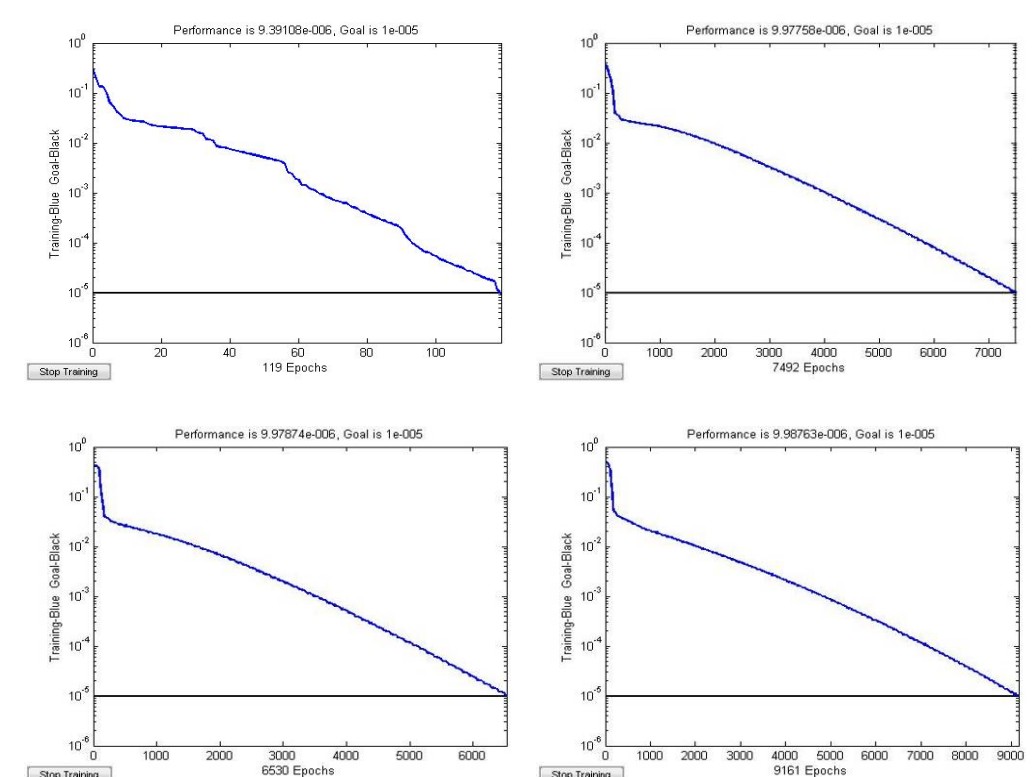

**Fig.8**





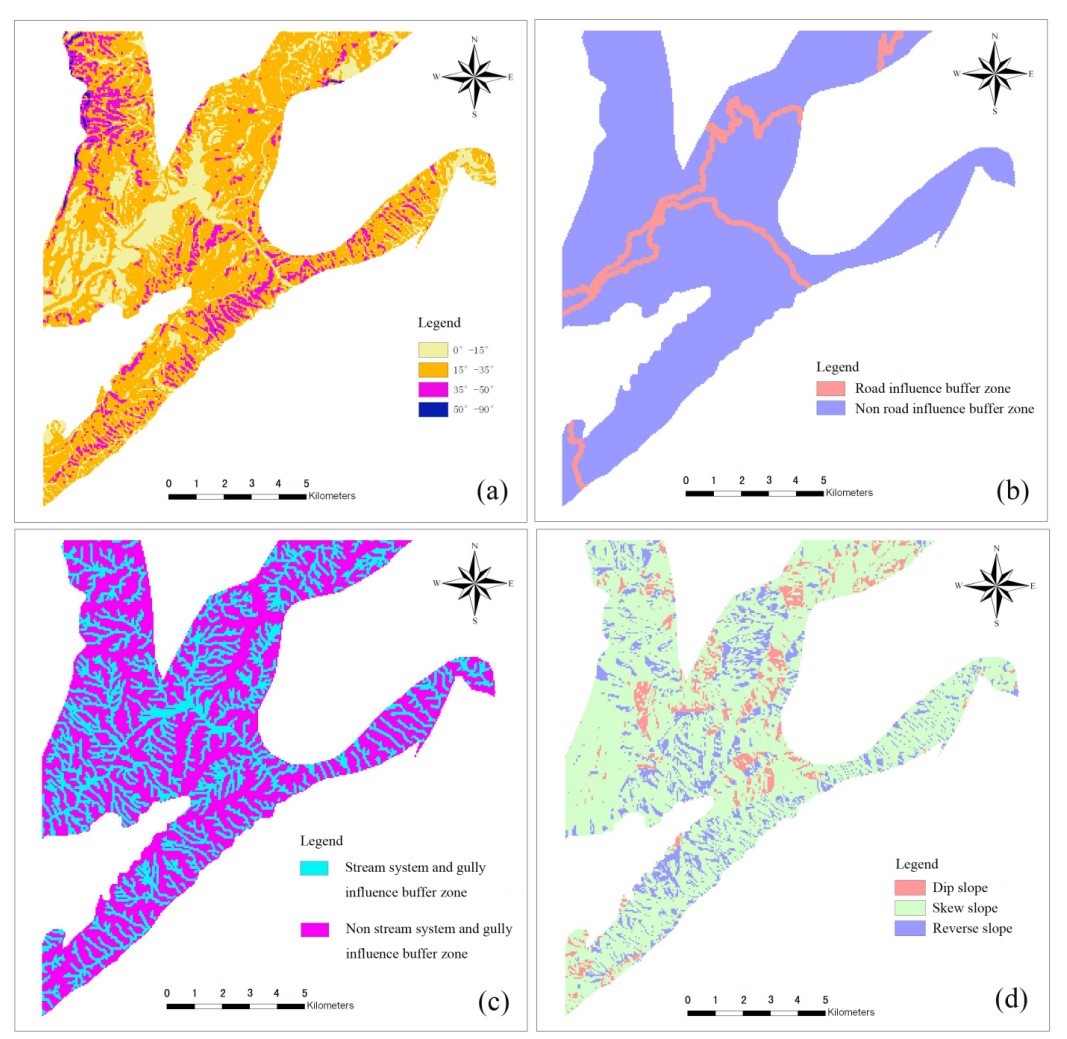

**Fig.9**



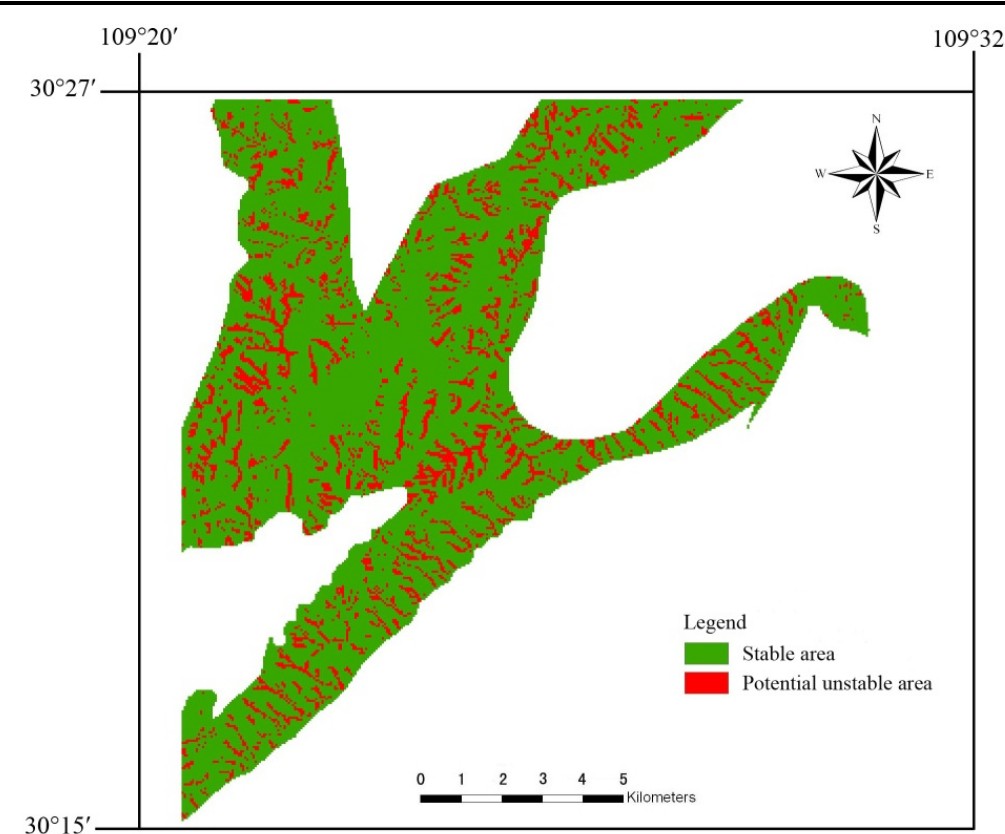

**Fig.10**




**Table 1 Three different slope structures and the corresponding angle ranges**

| | angle (θ) range | |
|---|---|---|
| dip slope | $0° < |θ| < 20°$;$340° < |θ| < 360°$ | |
| reverse slope | $160° < |θ| < 180°$;$180° < |θ| < 200°$ | |
| skew slope | $20° < |θ| < 160°$;$200° < |θ| < 340°$ | |

"θ" is the angle between stratum dip direction and slope surface dip direction.



**Table 2 Normalized sample data**

| Sample No | Road effect buffer | Slope angle | Slope structure type | Stream and gully effect buffer | Slope stability state | Sample No | Road effect buffer | Slope angle | Slope structure Type | Stream and gully effect buffer | Slope stability state |
|---|---|---|---|---|---|---|---|---|---|---|---|
| 1 | 0.1 | 0.26 | 0.9 | 0.9 | 1 | 19 | 0.9 | 0.34 | 0.5 | 0.9 | 1 |
| 2 | 0.1 | 0.55 | 0.9 | 0.1 | 0 | 20 | 0.1 | 0.37 | 0.9 | 0.1 | 0 |
| 3 | 0.1 | 0.29 | 0.9 | 0.1 | 1 | 21 | 0.1 | 0.42 | 0.5 | 0.9 | 1 |
| 4 | 0.1 | 0.23 | 0.9 | 0.9 | 1 | 22 | 0.9 | 0.1 | 0.1 | 0.9 | 0 |
| 5 | 0.1 | 0.58 | 0.1 | 0.9 | 0 | 23 | 0.1 | 0.28 | 0.5 | 0.9 | 1 |
| 6 | 0.9 | 0.37 | 0.9 | 0.1 | 1 | 24 | 0.1 | 0.66 | 0.9 | 0.9 | 1 |
| 7 | 0.1 | 0.55 | 0.5 | 0.9 | 0 | 25 | 0.1 | 0.42 | 0.5 | 0.1 | 1 |
| 8 | 0.1 | 0.74 | 0.9 | 0.9 | 0 | 26 | 0.1 | 0.34 | 0.9 | 0.9 | 1 |
| 9 | 0.1 | 0.34 | 0.1 | 0.1 | 0 | 27 | 0.1 | 0.82 | 0.9 | 0.9 | 0 |
| 10 | 0.9 | 0.42 | 0.9 | 0.9 | 1 | 28 | 0.1 | 0.45 | 0.9 | 0.9 | 1 |
| 11 | 0.9 | 0.58 | 0.5 | 0.1 | 0 | 29 | 0.9 | 0.16 | 0.5 | 0.1 | 0 |
| 12 | 0.9 | 0.26 | 0.1 | 0.1 | 0 | 30 | 0.1 | 0.39 | 0.5 | 0.9 | 1 |
| 13 | 0.9 | 0.1 | 0.9 | 0.9 | 0 | 31 | 0.1 | 0.36 | 0.9 | 0.9 | 1 |
| 14 | 0.1 | 0.34 | 0.9 | 0.9 | 1 | 32 | 0.9 | 0.66 | 0.9 | 0.9 | 1 |
| 15 | 0.9 | 0.5 | 0.9 | 0.9 | 1 | 33 | 0.9 | 0.5 | 0.9 | 0.9 | 1 |
| 16 | 0.1 | 0.9 | 0.9 | 0.1 | 0 | 34 | 0.1 | 0.15 | 0.9 | 0.9 | 0 |
| 17 | 0.1 | 0.61 | 0.9 | 0.9 | 0 | 35 | 0.9 | 0.39 | 0.5 | 0.9 | 1 |
| 18 | 0.9 | 0.42 | 0.9 | 0.9 | 1 | | | | | | |





**Table 3 Predictive ability test result of the neural network**

| Sample number | Predicted Value | Actual Value | Absolute Error |
|---|---|---|---|
| 26 | 0.99905 | 1 | 0.0009489 |
| 27 | 1.82e-06 | 0 | -1.82e-06 |
| 28 | 2.2017e-005 | 1 | 0.99998 |
| 29 | 0.014818 | 0 | -0.014818 |
| 30 | 0.99992 | 1 | 7.91e-05 |
| 31 | 0.99033 | 1 | 0.0096745 |
| 32 | 0.99999 | 1 | 5.56e-06 |
| 33 | 1 | 1 | 1.18e-09 |
| 34 | 1 | 0 | -1 |
| 35 | 0.99754 | 1 | 0.0024614 |



**Table 4 Comparison among the results of intelligent prediction, remote sensing and field investigation**

| Intelligent prediction results | Judgment by Remote sensing data | Judgment by filed investigation |
|---|---|---|
| | | |
| | | |