# Peer review of "Assessment of shallow landslide susceptibility using an artificial neural network in Enshi region, China"

_Natural Hazards and Earth System Sciences, 2017_

## Referee Comment (RC1) · Anonymous Referee #1 · 12 Jun 2017

The manuscript has very limited contribution to the field since many papers have been published in the last two decades on the use of neural nets in landslide susceptibility mapping. Authors should explain how they choose the parameters (learning rate, momentum, initial weight range) and why? For example when you choose 10.000 iterations in training stage, the network is high likely to overfit the data that is extremely limited in size.

Another important problem is related to the sample size, which I believe is too limited, for the designed network (4-9-1). This network has totally 45 links. The number of training samples employed at the learning stage has a significant impact on the performance of any classifier. This issue is perhaps more important for neural networks than for conventional statistical classifiers since their performance is totally dependent

upon the characteristics of the training data presented. Although the size of the training data is of considerable importance, the characteristics and the distributions of the data as well as the sampling strategy used are crucial. the quality and the quantity of the training samples are crucially important for a successful neural network application. Whilst too few training samples are not sufficient for neural networks to derive the characteristics of the classes, the use of too large a number of training samples may cause networks to overfit to the data, as well as requiring more time for learning.

On page 4 lines 54 and 61 the surname of the author (Pourghasemi) was written incorrectly. On page 6 line 87 "compare with... some different attempts" can be replaced as "compared with .... some attempts".

Conclusion section is too short and includes general information.

Provide more recent papers in reference list in the use of ANNs and its performance comparison to machine learning methods.

———————————————————

---

## Referee Comment (RC2) · Anonymous Referee #2 · 23 Jun 2017

GENERAL COMMENTS

In this paper the authors implemented an artificial neural network (ANN) approach for analyzing landslide susceptibility within a certain geological unit in a study area in China, taking into account four different factors related to the landslide mechanism, and endangered slopes rather than already failed slopes as positive inputs for the model.

As I see it, the overall quality of the paper is at this point not adequate for publication in NHESS, and I suggest major revisions.

First of all, the English language needs improvement, with many grammatical errors and imprecise wording, and the manuscript should be completely revised by a more proficient English user. Secondly, the description of the methodology is not straightfor-

ward and it is hard to follow what exactly the authors did and where the data comes from. The discussion of the results and the performance evaluation are very short and not going into detail enough or documented with adequate performance evaluation methods. Finally, it is hard for me to see the scientific contribution of the work. There are already many papers using artificial networks for landslide susceptibility analysis. The idea of investigating only a single geological unit is not new to me, neither is the implementation of the catchment area as input. The principle of using endangered slopes as input rather than occurred landslides is interesting, but not explained enough, particularly regarding the mapping and generation of such an inventory, as well as issues arising from its completeness or incompleteness. The implications of the results for advancing the understanding of natural hazards are not pointed out enough.

SPECIFIC COMMENTS

Line 64: unclear what the authors mean with "intelligent method".

Line 78: "valuation index system" is unclear.

Line 80: "empirical" instead of "experiential"?

Lines 80 to 81: The role of expert knowledge and experience: Is not one of the major points of data driven landslide susceptibility analysis to move away from expert knowledge towards being more objective?

Lines 194 and following section 3.1.2: Here it is hard to understand what exactly was done without having the information about the input data, such as DEM and grid size, which is given later in the results chapter 4.

Line 210: where is 50 m defined as the minimum grid size for statistical analyses? Is that in any other publication? It is unclear if it was used here. Later the DEM is referred to with a 25 m grid size. This is very confusing.

Lines 212 and following: Where does the SC come from? Is it quoted from another publication?

Lines 227 to 232: It is unclear how the structural geological model was generated and what kind of input data was used for it. This is a nontrivial issue and it needs to be clarified.

Line 237: "...avoid the interference of human factors." Here the authors contradict the statement from lines 80 to 81.

Line 272: Can the authors give more details about the "traingdx" function?

Lines 289 to 294: Which sampling units were used for the calculations? Grid cells? Slope units? How many grid cells correspond to the respective samples? Similar Line 314: How many grid cells/sampling units do the different samples correspond to? This is crucial for the reader to understand the sample size and the significance of the analysis.

Line 317 and following section 4.1: This section should be moved to the methods chapter.

Lines 359 to 362: This statement and the general performance of the model should be better quantified with adequate indices.

Line 249: I do not understand in which regard the assessment was dynamic.

Quality of the results and discussion in general: The presented results are of only little significance. Neither do the authors point out a way of optimizing the model, e.g. by changing the architecture of the ANN or set of input variables, or using committees, nor do they go into depth when interpreting the results regarding the very particular landslide mechanism or point out weaknesses of the approach. Also, the figures and tables supporting the presentation of the results are not adequately explained and discussed in the text. They cannot be considered self-explaining.

Lines 384 to 392: points 1 and 3 are not new to me, point 2 not explained enough, so I have doubts regarding the novelty of this research. Moreover, the implications of the achievements and possible practical implementations are not pointed out.

Fig. 3: What are the profile lines in the figure? Are they of any relevance? Otherwise they should be omitted.

TECHNICAL CORRECTIONS

Many grammatical errors all over the manuscript that have to be fixed.

Lines 44 and 46: Pradhan and Lee 2010 should be distinguished, there are two publications in the reference section.

Lines 54 and 61: Spelling of author Pourghasemi incorrect.

---

## Author Comment (AC1) · 25 Aug 2017

Dear Referee and Editor:

Thank you for the valuable suggestions. We have carefully read through the comments, and our responses to the referee's questions are listed below. We greatly appreciate your time and efforts to improve our manuscript for further revision and publication.

1. The manuscript has very limited contribution to the field since many papers have been published in the last two decades on the use of neural nets in landslide susceptibility mapping. Authors should explain how they choose the parameters (learning rate, momentum, initial weight range) and why? For example when you choose 10.000 iterations in training stage, the network is high likely to overfit the data that is extremely

limited in size. Re: Compare with existing studies, some new attempts have been carried out in this paper: (1) this research focused on the distribution of unstable slope zones rather than the existing landslides, since the unstable slopes are more dangerous; (2) this research predicted the unstable slope distribution only in Silurian stratum so as to avoid the interference due to differences in slope (in different stratum) failure mechanisms; (3) a "slope structure thematic map" was taken into account to better represent the especial slope failure mechanism in Silurian stratum; (4) replaced the temporal variable of rainfall into a static, spatial variable termed "catchment area" to better act as an influencing factor during the landslide susceptibility. The research results can provide useful guidance for both landslide susceptibility assessment and land planning processes. In the chapter 3.2.2, the paper explained the decision process of many parameters like "the number of neurons for the input and output layers", "the number of hidden layers", "the number of neurons in the hidden layer", "the network training function". And In the BP neural network, "the initial network weights and thresholds" were given random values in the acceptable range, based on the theory that an initial value which is not too large has little impact on the overall performance of the network, while a smaller initial range is more conducive to uniformly random initial weights (Freeman, 1993). The author has carried out a lot of tentative calculations, so as to compare the ability of different training functions (traingd, traingdm, traingdx, trainlm etc.). After choosing "traingdx" as the training function of the neural network, the author also carried out a lot of tentative calculations to determine the number of training iteration (10000 times), so as to ensure that the network can achieve the training goal and converge.

2. Another important problem is related to the sample size, which I believe is too limited, for the designed network (4-9-1). This network has totally 45 links. The number of training samples employed at the learning stage has a significant impact on the performance of any classifier. This issue is perhaps more important for neural networks than for conventional statistical classifiers since their performance is totally dependent upon the characteristics of the training data presented. Although the size of the training data is of considerable importance, the characteristics and the distributions of the data as well as the sampling strategy used are crucial. The quality and the quantity of the training samples are crucially important for a successful neural network application. Whilst too few training samples are not sufficient for neural networks to derive the characteristics of the classes, the use of too large a number of training samples may cause networks to overfit to the data, as well as requiring more time for learning. Re: The final forecasting area of this research is about 103 km2, which is the range of Silurian stratum in Enshi region. In this area, 35 stable and unstable slopes in Silurian stratum in Enshi region were chosen as the sample data, the recognition and mapping work have been carried out by geomorphological field survey. All the 35 chosen samples fitted the failure mechanism of landslides in Silurian stratum as discussed in chapter 2.2.2 and 2.2.3 and covered as much as possible the different combinations of various factors to improve the forecasting ability of the network. So that the neural network trained by these samples, can effectively approximate the inherent law of the samples by studying and remembering the known samples, then carry out an associated forecast according to the memory.

3. On page 4 lines 54 and 61 the surname of the author (Pourghasemi) was written incorrectly. On page 6 line 87 "compare with... some different attempts" can be replaced as "compared with .... some attempts". Re: The author will correct this mistake, and also check the whole manuscript to avoid similar mistakes.

4. Conclusion section is too short and includes general information. Re: The author will further revise the conclusion chapter: adding more discussion about the research significance and novelty, and simplify the general information.

5. Provide more recent papers in reference list in the use of ANNs and its performance comparison to machine learning methods. Re: The author will supply more recent references concerning the use of ANNs and its performance comparison to machine learning methods, and will also discuss these research results in the proper chapter of the paper.

---

## Author Comment (AC2) · 25 Aug 2017

Dear Referee and Editor:

Thank you for the valuable suggestions. We have carefully read through the comments, and our responses to the referee's questions are listed below. We greatly appreciate your time and efforts to improve our manuscript for further revision and publication..

GENERAL COMMENTS

In this paper the authors implemented an artificial neural network (ANN) approach for analyzing landslide susceptibility within a certain geological unit in a study area in China, taking into account four different factors related to the landslide mechanism, and endangered slopes rather than already failed slopes as positive inputs for the model.

[Figure]

As I see it, the overall quality of the paper is at this point not adequate for publication in NHESS, and I suggest major revisions.

1. First of all, the English language needs improvement, with many grammatical errors and imprecise wording, and the manuscript should be completely revised by a more proficient English user.

Re: The language of this paper was revised by "American Journal Experts" before submitting. And the author will ask either an English user or another language service organization to completely revise the language of this paper before submitting the revised manuscript.

2. Secondly, the description of the methodology is not straightforward and it is hard to follow what exactly the authors did and where the data comes from. The discussion of the results and the performance evaluation are very short and not going into detail enough or documented with adequate performance evaluation methods.

Re: In the "method" section, the author mainly introduced two aspects in turn. One is how to establish the index system, including: index filtration, index quantification, index weights determination and other key issues; the second is how to establish the artificial neural network model, including: neural network type selection, establishment of BP neural network model, preparation of sample data. In the discussion section, the paper showed the prediction results, and the prediction results were validated by means of field investigation and remote sensing interpretation. The paper finally analyzed the reliability and applicability of the artificial intelligence prediction method established for the prediction of unstable slopes.

3. Finally, it is hard for me to see the scientific contribution of the work. There are already many papers using artificial networks for landslide susceptibility analysis. The idea of investigating only a single geological unit is not new to me, neither is the implementation of the catchment area as input. The principle of using endangered slopes as input rather than occurred landslides is interesting, but not explained enough, particularly regarding the mapping and generation of such an inventory, as well as issues arising from its completeness or incompleteness. The implications of the results for advancing the understanding of natural hazards are not pointed out enough.

Re: Compare with existing studies, some new attempts have been carried out in this paper: (1) this research focused on the distribution of unstable slope zones rather than the existing landslides, since the unstable slopes are more dangerous; (2) this research predicted the unstable slope distribution only in Silurian stratum so as to avoid the interference due to differences in slope (in different stratum) failure mechanisms; (3) a "slope structure thematic map" was taken into account to better represent the especial slope failure mechanism in Silurian stratum; (4) replaced the temporal variable of rainfall into a static, spatial variable termed "catchment area" to better act as an influencing factor during the landslide susceptibility. The research results can provide useful guidance for both landslide susceptibility assessment and land planning processes.

SPECIFIC COMMENTS

4. Line 64: unclear what the authors mean with "intelligent method".

Re: "intelligent method" means the artificial neural network method which was used in this manuscript.

5. Line 78: "valuation index system" is unclear.

Re: "valuation index system" means the index system which is used for evaluate the susceptibility of unstable slopes, and it also should obey the failure mechanism of the research landslide.

6. Line 80: "empirical" instead of "experiential"?

Re: the author will replace this word in the manuscript.

7. Lines 80 to 81: The role of expert knowledge and experience: Is not one of the major points of data driven landslide susceptibility analysis to move away from expert

knowledge towards being more objective?

Re: in this part, the author trying to explain that, in the process of the neural network prediction, the network have to study the landslide failure mechanism based on the sample data which was come from the expert knowledge and experience.

8. Lines 194 and following section 3.1.2: Here it is hard to understand what exactly was done without having the information about the input data, such as DEM and grid size, which is given later in the results chapter 4.

Re: in this section, the paper mainly discussed the establishment of index system, filtration of index and so on. After the index system was confirmed, the information about DEM, grid size and thematic layer were given in the "Results and Discussion" part. The author will move this important information to the "Method" part.

9. Line 210: where is 50 m defined as the minimum grid size for statistical analyses? Is that in any other publication? It is unclear if it was used here. Later the DEM is referred to with a 25 m grid size. This is very confusing.

Re: the manuscript took 25 m as the smallest statistic cell, in road influence buffer zone analysis, stream system and gully influence buffer zone analysis and final grid size design. In line 210, it was a mistake.

10. Lines 212 and following: Where does the SC come from? Is it quoted from another publication?

Re: the author will add the relevant reference in the "Reference" part.

11. Lines 227 to 232: It is unclear how the structural geological model was generated and what kind of input data was used for it. This is a nontrivial issue and it needs to be clarified.

Re: First, the stratum dip directions were divided into 8 ranges: 0°-45°, 45°-90°, 90°-135°, 135°-180°, 180°-225°, 225°-270°, 270°-315°, 315°-360°, and the stratum dip

direction distribution map can be obtained. Second, the slope surface dip direction distribution map can be calculated by ArcGIS based on DEM data. Third, the angle between the slope surface dip direction and the stratum dip direction can be obtained by subtracting the superposition calculation from the above two layers. Finally, the slope structure can be defined (dip slope, reverse slope, skew slope) in the study area according to the Table 1 in the manuscript.

12. Line 237: ". . .avoid the interference of human factors." Here the authors contradict the statement from lines 80 to 81.

Re: The statement from 80 to 81 means the neural network can simulate the learning and judging process of an expert. And the line 237 means that, after mastering the mechanism of slope failure from sample study, the neural network can distribute the weight automatically, so as to avoid the different judgments from different experts.

13. Line 272: Can the authors give more details about the "traingdx" function?

Re: the author will supply the detailed information about "traingdx" function, and also the screening process of the different functions.

14. Lines 289 to 294: Which sampling units were used for the calculations? Grid cells? Slope units? How many grid cells correspond to the respective samples? Similar Line 314: How many grid cells/sampling units do the different samples correspond to? This is crucial for the reader to understand the sample size and the significance of the analysis.

Re: in this manuscript, considering the research area and scale, the Grid cells which set as 25 m × 25 m were used for the calculation. In the revised manuscript, the author will supply the exact grid numbers about the respective samples during all the training and testing process.

15. Line 317 and following section 4.1: This section should be moved to the methods chapter.

Re: in the revised manuscript, the author will adjust the section 4.1 to "methods" chapter.

16. Lines 359 to 362: This statement and the general performance of the model should be better quantified with adequate indices.

Re: the author will supply more detailed and quantified information about the remote sensing data and field investigation situation, so as to better assess the forecast results of the neural network.

17. Line 349: I do not understand in which regard the assessment was dynamic.

Re: Because of the repeated learning model of the neural network, the weight of the model is determined by the dynamic adjustment of the model itself. In the course of constant adjustment, the model can approach the best learning effect and meet the characteristics of the studied landslide.

18. Quality of the results and discussion in general: The presented results are of only little significance. Neither do the authors point out a way of optimizing the model, e.g. by changing the architecture of the ANN or set of input variables, or using committees, nor do they go into depth when interpreting the results regarding the very particular landslide mechanism or point out weaknesses of the approach. Also, the figures and tables supporting the presentation of the results are not adequately explained and discussed in the text. They cannot be considered self-explaining.

Re: The author will further revised the "results and discussion" part, supplying more detailed information about the model optimization, discussing more about how the neural network model fitting the studied landslide failure mechanism, explaining more about the figures and tables that how they supporting the research results.

19. Lines 384 to 392: points 1 and 3 are not new to me, point 2 not explained enough, so I have doubts regarding the novelty of this research. Moreover, the implications of the achievements and possible practical implementations are not pointed out.

Re: Compare with existing studies, some new attempts have been carried out in this paper: (1) the research focused on the distribution of unstable slope zones rather than the existing landslides, since the unstable slopes are more dangerous; (2) this research predicted the unstable slope distribution only in Silurian stratum so as to avoid the interference due to differences in slope (in different stratum) failure mechanisms; (3) a "slope structure thematic map" was taken into account to better represent the especial slope failure mechanism in Silurian stratum; (4) replaced the temporal variable of rainfall into a static, spatial variable termed "catchment area" to better act as an influencing factor during the landslide susceptibility. In the "conclusion" part, the author will supply more detailed information about the implications of the achievements and possible practical implementations

20. Fig. 3: What are the profile lines in the figure? Are they of any relevance? Otherwise they should be omitted.

Re: the author will delete the profiles in Fig. 3.

TECHNICAL CORRECTIONS

21. Many grammatical errors all over the manuscript that have to be fixed. Lines 44 and 46: Pradhan and Lee 2010 should be distinguished, there are two publications in the reference section. Lines 54 and 61: Spelling of author Pourghasemi incorrect.

Re: the grammar of the manuscript has been optimized by "American Journal Experts" before submitting. The author will ask another organization to further edit and optimize the grammar in the revised manuscript. And the author will also check the reference and spelling problems during the whole manuscript.